# Assessment of an Ultrasound-Guided Rectus Sheath Block in Foals: A Cadaveric Study

**DOI:** 10.3390/ani13233600

**Published:** 2023-11-21

**Authors:** Álvaro Jesús Gutiérrez Bautista, Franz Josef Söbbeler, Rüdiger Koch, Jaime Viscasillas, Sabine Kästner

**Affiliations:** 1Small Animal Clinic, University of Veterinary Medicine Hanover, Foundation, Bünteweg 9, 30559 Hannover, Germany; 2Institute of Anatomy, University of Veterinary Medicine Hanover, Foundation, Bischofsholer Damm 15, 30173 Hanover, Germany; ruediger.koch@tiho-hannover.de; 3Hospital Veterinario Anicura Valencia Sur, Av. de Picassent, 28, 46460 Valencia, Spain; 4Clinic for Horses, University of Veterinary Medicine Hannover, Foundation, Bünteweg 9, 30559 Hannover, Germany; sabine.kaestner@tiho-hannover.de

**Keywords:** rectus sheath block, foal, ultrasound-guided regional anaesthesia, spinal nerves, umbilical surgery

## Abstract

**Simple Summary:**

Foals are commonly subjected to surgeries related to umbilical pathologies. Due to side effects of systemic analgesics and the immaturity of their organ systems, it is challenging to provide adequate analgesia. It is in this context that the role of regional anaesthesia allows for the provision of good quality analgesic coverage, reducing the potential complications associated with systemic analgesic drugs such as opioids. Ultrasound-guided locoregional techniques are commonly employed in both human and veterinary medicine to provide accurate, safe, and effective analgesia in the regions of interest. The rectus sheath block provides adequate analgesia in procedures involving the umbilical area in humans and has been described in other species, including dogs, pigs, calves, and lately in adult horses. The aim of this study was to describe an ultrasound-guided periumbilical rectus sheath block in foal cadavers and evaluate the spread of the contrast media injected. The results obtained indicated that this approach could provide analgesia around the umbilical area due to the nerves stained. However, further studies evaluating the clinical efficacy of the rectus sheath block are necessary.

**Abstract:**

In neonatal equines, pathologies involving umbilical structures are an important cause of morbidity, and surgical removal of urachal remnants is a common procedure in clinical practice. Surgery involving the ventral abdominal wall can cause substantial pain, leading to complications and prolonged recovery. The objectives of this study were to describe a two-point bilateral ultrasound-guided rectus sheath block at the level of the umbilicus and to evaluate the extent of dye distribution in foal cadavers. Ten foal cadavers were included in the study, in which a bilateral two-point ultrasound-guided rectus sheath block was performed—one injection 5 cm cranially and a second one 5 cm caudally to the umbilicus. The injectate consisted of a mixture of iodinated contrast medium and blue dye at a volume of 0.25 mL kg^−1^ per injection point (total 1 mL kg^−1^). After the injection, computer tomography and subsequent dissection of the ventral abdominal wall were performed. The extension of the contrast medium, the number of stained nerves, and contamination of the abdominal cavity were evaluated. The cranio-caudal extension of the contrast ranged from 0.8 to 1.4 cm per milliliter of injectate. The most commonly stained ventral branches of spinal nerves were thoracic (Th) nerves 16, 17, and 18 (95%, 85%, and 80% of the nerves, respectively). Abdominal contamination was found in four animals. The results suggest that the block could provide periumbilical analgesia. Further studies with different volumes of injectate and living animals are warranted.

## 1. Introduction

A laparotomy is an invasive procedure frequently performed in horses. It is associated with substantial somatic pain and visceral hyperalgesia necessitating early and aggressive treatment. Early treatment of pain results in improved recovery and reduced hospitalization time [1]. Discomfort in those patients can increase the risk of the development of postoperative colic. Persistent pain after surgery can result in sympathetic overstimulation which can lead to altered gastrointestinal motility [2,3,4,5]. Systemic analgesics like opioids and alpha-2 adrenoceptor agonists are commonly employed to treat pain, although they can cause side effects like cardiovascular depression or gastrointestinal transit delay and ileus [6]. In this context, the inclusion of locoregional techniques could be beneficial to provide effective analgesia while reducing systemic side effects.

Locoregional techniques have gained popularity in both human and veterinary medicine to treat perioperative pain [7,8]. The rectus sheath block was first described in people by Schleich in 1899 [9] and it was initially used for abdominal wall muscle relaxation and analgesia during a midline laparotomy by blocking the terminal branches of the thoracolumbar nerves. It is indicated for laparoscopic surgery, periumbilical incisions, umbilical and paraumbilical hernia repair, abdominoplasty, and open radical cystectomy [9]. In the equine neonate, pathologies involving the umbilical structures are an important cause of morbidity and surgical removal of the urachal remnants is a common procedure in clinical practice [10]. The immaturity of the different organ systems of the neonatal foal makes them more susceptible to the pharmacologic side effects of systemic drugs [11]. This makes the inclusion of locoregional techniques into balanced anaesthetic protocols interesting in order to reduce the consumption of analgesic agents, as reported in human medicine [12].

In horses, the lateral and ventral abdominal wall is innervated by the ventral branches of the 10th to 18th thoracic (Th) spinal nerves. They run within the fascial plane between the *m. transversus abdominis* and the *m. obliquus internus abdominis.* In the most ventral aspect of the abdomen at the level of the internal rectus sheath, they pierce the internal face of the *m. rectus abdominis* and innervate the muscle and skin in the vicinity of the *linea alba* [13]. They carry motor fibers for the *m. rectus abdominis*, sensory fibers for the parietal peritoneum, and close to the *linea alba,* they give rise to the short ventral cutaneous branches [14]. Interfascial blocks to provide abdominal wall analgesia have been described in cadaveric studies in horses [15]. The rectus sheath block has been described in cadaveric studies in dogs, pigs, and calves with promising results [7,16,17,18]. A study evaluated a two-point rectus sheath block in the middle point between the xyphoid process and the umbilicus. The authors reported a significant increase in mechanical nociceptive threshold in adult horses [19]. To the author’s knowledge, the literature evaluating rectus sheath block in equines is scarce and there is no cadaveric study describing the distribution after periumbilical injection in foals. 

The aims of this study were to describe a two-point bilateral ultrasound-guided rectus sheath block at the level of the umbilicus, and to evaluate the extent of the dye achieved using computed tomography and the staining of the ventral branches of the thoracic spinal nerves through gross anatomical dissection in foal carcasses. We hypothesized that an injection at the point described would reach the ventral branches of the thoracic spinal nerves within the interfascial plane. 

## 2. Materials and Methods

Foal carcasses were obtained from the Clinic for Horses of the University of Veterinary Medicine Hannover, Foundation and were euthanized for reasons unrelated to the study. Informed written consent from the owners was obtained for all animals. No animal use protocol or ethical approval was required for the use of foal cadavers according to the university policies.

The study was divided into two parts. Part I involved an examination of the anatomy of the abdominal wall, with a specific focus on the ventral aspect and structures closely related to the rectus sheath muscle and the terminal branches of the intercostal nerves (*nn. intercostales, rr. cutanei laterales*). For this purpose, 1 foal cadaver weighing 40 kg was used. Part II consisted of performance and evaluation of the ultrasound-guided rectus sheath block in fresh foal cadavers.

### 2.1. Part I: Anatomical Examination

The foal was positioned in lateral recumbency, and a meticulous dissection of the lateral and ventral abdominal wall was conducted. The skin, *mm. cutaneous trunci*, *mm obliquus externus*, and *mm. obliquus internus* were retracted to expose the surface of the *mm. transversus abdominis*, *mm. rectus abdominis*, and the ventral branches of the last thoracic and first lumbar spinal nerves. The anatomical evaluation and dissection were performed by a veterinary anatomist (RK), with the veterinary anaesthetist (AG) who performed the blocks in Part II.

### 2.2. Part II: Rectus Sheath Block Trials

Ten fresh foal cadavers with an average weight of 37 ± 5 kg (mean ± SD) were included in the study. The number of animals included was based on a previous study performed in foal cadavers [15]. The study was performed within 24 h after euthanasia. Before the experiment, the cadavers were stored in a cold room. Inclusion criteria included foals with no history of trauma and/or abdominal surgery and immediate availability for the study after euthanasia. 

Four injections were performed on each abdomen (one cranial and one caudal to the umbilicus, on each hemiabdomen), with a volume of injectate of 0.25 mL kg^−1^ in each one (total of 1 mL kg^−1^ for each animal). The injectate consisted of a mixture of 1/5 iodate contrast medium (Imeron^®^ 300 mg mL^−1^, Bracco Imaging Deutschland GmbH, Konstanz, Germany) plus 4/5 of blue dye (Tissue Marking Dyes, Blue, Cancer Diagnostics, Inc., Durham, NC, USA). The original dye was diluted (1 mL of dye in 250 mL of NaCl 0.9%) to cause no marked changes in viscosity of the injectate. The determination of the injectate volume was based on preliminary data from a study (unpublished) conducted on cadavers not included in the results.

### 2.3. Ultrasound-Guided Rectus Sheath Block

The cadavers were placed in dorsal recumbency, and their limbs were secured with ropes to stabilize them. The area around the umbilicus was clipped, leaving a 15 cm margin in the cranial, caudal, and bilateral directions. A linear 12 MHz ultrasound probe (Ultrasound Transducer 12L-RS, GE Medical Systems, China Co., Ltd., Wuxi, China) connected to an ultrasound machine (Logic V2; GE Medical Systems, China Co., Ltd.) was used. Initially, the linear array ultrasound transducer was positioned in a transverse orientation just cranial to the umbilicus to identify the rest of the *urachus*, *linea alba*, and transversal window of the *mm. rectus abdominis*. Subsequently, the transducer was advanced 5 cm cranially and then laterally until the lateral border of the *mm. rectus abdominis*, which had a triangular shape, was located. Two hyperechoic railway-like lines deep within it were identified as the posterior rectus sheath and fascia transversalis (6). The injection point was determined to be between the rectus abdominis and the posterior sheath, positioned at the lateral third of the muscle (immediately before the nerves puncture and enter the muscle, as described in Part I). A 22-gauge, 75 mm Quincke needle (Spinal needle Quincke type point, BD S.A., Madrid, Spain), attached to an extension and a syringe pre-filled with the solution, was inserted in a medio-ventral to latero-dorsal direction with a 30-degree angle to the probe, using an “in-plane” technique. When the needle tip was believed to be in the target point, a small aliquot of the solution was injected to create a “pocket” in the interfascial plane, and subsequently, the rest of the volume was administered to cause hydrodissection of the plane. If the aliquot was not in the correct place, repositioning of the needle was performed. This process was repeated as described 5 cm caudal to the umbilicus and in the contralateral hemiabdomen (Appendix A). After the 4 first animals, where intraperitoneal contamination was detected, the injection technique was slightly modified. Administration of the injectate was started before the desired point was reached while advancing the needle, so that a “pocket” was created to avoid intraperitoneal contamination.

### 2.4. Evaluation of the Quality of the US Image and Needle Visualization

The classifications below (Table 1 and Table 2), described in a similar study in dog’s cadavers [7], was employed:

### 2.5. Imaging Study and Dissection

After the injection, a computed tomography (CT) scan of the abdomen was performed and evaluated by a different operator (FJS). Image acquisition for all the cadavers was conducted using a 16-slice multiple-detector CT scanner (BrillianceTM CT, Philips Medical Systems, Best, The Netherlands). The foals were placed in dorsal recumbency. The CT images were acquired using helical acquisition, with a slice thickness of 0.9 mm, a helical pitch of 0.567, a tube current of 354 mAs, a tube potential of 140 kVp, a matrix size of 1024 × 1024, and a reconstruction algorithm for both soft tissue and bone. The images were reviewed using a bone window (window level 300, window width 2800). The following CT features were documented: intraperitoneal and intramuscular contrast contamination, the relative cranio-caudal extent achieved of the contrast (in centimeters per milliliter of injectate), and its relationship with the last thoracic, lumbar, and sacral vertebrae.

Subsequently, the foals were transported to a dissection room and positioned in lateral recumbency. To maintain the anatomical reference, the umbilicus was marked with a metal clamp. A skin incision was made with a semicircular shape from the xiphoid, along the costal arch to the iliac crest. The incision was then extended directly to the ventral abdominal midline. A meticulous dissection followed, involving the layers of the abdominal wall in the following sequence: *cutis*, *mm. cutaneus*, *mm. obliquus externus*, and *mm. obliquus internus*. The *mm. rectus abdominis* was then isolated from *mm. transversus abdominis*, exposing the thoracic nerves as they descended over the *mm. transversus abdominis* and identifying the point where they entered the internal face of the *mm. rectus abdominis*. The first nerve caudal to the umbilicus was traced to its origin in the intervertebral foramen for accurate identification. Nerve staining was classified using a previously employed scale (Table 3) [7], and the intra-abdominal location of the dye was also documented. Results are presented as frequency (n/n) and percentage (%) of nerves stained from the total amount of nerves identified.

## 3. Results

### 3.1. Ultrasound Injection and CT Scan

The quality of the ultrasound image was excellent in all cases, with clear visibility of the anatomy and reference points. Needle visualization was excellent in all cases except one, where it was classified as good. The CT scan results are detailed in Table 4. Figure 1 illustrates the extent of the contrast in relation to the lumbar and sacral vertebrae. The CT scan results for the second animal (RS2) had to be excluded due to technical issues with the scanner.

### 3.2. Dissection

Table 5 describes the degree and frequency of staining of the nerves of interest examined in the dissection. Appendix A shows the dissection and identification of the stained nerves in one foal. 

The location of the umbilicus in relation to the spinal nerves is shown in Table 6.

## 4. Discussion

This is the first study that describes a four-point periumbilical ultrasound-guided rectus sheath block in foals with a total volume of 1 mL kg^−1^ divided into four equal aliquots. The aim was to evaluate the extent of dye and to quantify the number of spinal nerves reached. The high percentage of stained nerves around the area of interest suggests that the technique could provide effective pain relief for umbilical procedures. Clear sonographic images of the anatomy and the needle were obtained during the injection. The hydrodissection was observed during the administration of the stain, confirming the correct placement of the needle, similar to the results obtained by other authors [17,19]. 

In a systematic review and meta-analysis in 2014, epidural analgesia was considered the gold-standard technique in human medicine for abdominal surgery, due to the strong analgesic effect on the corresponding dermatomes and to the reduction in mortality and morbidity outcomes [20]. In veterinary medicine, epidural or intrathecal anaesthesia is also performed for abdominal procedures, especially in small animals [21]. In equine patients, epidural anaesthesia can be performed similarly, however potential important complications like hypotension, postoperative ataxia, or muscle weakness can be fatal in equine patients. This makes the quest for alternative peripheral locoregional analgesic methods even more important [22].

There are several cadaveric studies evaluating locoregional techniques for the abdominal wall in horses. Baldo et al. [23] performed a one-point injection transversus abdominis plane (TAP) block in the midpoint between the iliac crest and the most caudal extent of the last rib, injecting 0.5 mL kg^−1^ per hemiabdomen. Although the technique successfully reached some of the ventral branches of the nerves involved in the innervation of the abdominal wall (mostly Th16, Th17, and Th18), a dorsal spread of the dye was observed with involvement of major psoas and quadratus lumborum muscles. This could lead to femoral nerve blockade and complications during recovery from general anaesthesia [23]. This complication has already been reported in human medicine [24] and it should be noted that the consequences in equine medicine can be much more serious. Approaching the spinal nerves in a more distal position, like in the rectus sheath, eliminates this risk, as shown in our results. The authors suggested to add an additional injection site as described in other species to improve the spreading and reach additional cranial nerves [25]. To achieve this, we decided to use a two-point approach, but our results regarding the spread of the injectate are comparable to the ones obtained by Baldo et al. in 2018. On the other hand, the aim of this study was to reach the nerves involved in procedures involving the umbilical structures. The nerves more frequently stained were those involved in the innervation of such structures (Th16, Th17, and Th18). Similarly, in 2020, Küls et al. published a modified three-point TAP block in cadavers and in living ponies, successfully desensitizing dermatomes of the lateral and ventral abdominal wall from Th8 to Th18 with a total volume of 0.3 mL kg^−1^ per hemiabdomen [6]. Even though, with this approach, the authors managed to desensitize the area involved in an exploratory laparotomy, we decided to apply a two-point technique to provide analgesia for umbilical surgeries, with a smaller area desensitized, to avoid a third injection and its potential complications.

Foals are commonly presented for surgery of the umbilical area due to pathologies like umbilical remnant infection or umbilical hernia repair [10,26]. Administration of analgesic drugs before application of a nociceptive stimulus reduces postoperative pain and analgesic requirements, and may decrease the incidence of peripheral and central sensitization [27]. In humans, superior pain relief and faster postoperative recovery is achieved when preemptive regional anaesthetics are employed in comparison with opioid therapy [28]. Infiltration of local anaesthetics around the surgical area is a commonly employed technique for perioperative pain management [29]. However, the use of US-guided techniques presents several advantages when compared to blind techniques: visualization of the targeted area (nerve or fascial plane), need for smaller volumes of local anaesthetics because of proximity to the nerve, and avoiding puncture of neural and vascular structures and organ damage [30]. Rectus sheath block is used to provide effective pain relief for umbilical hernia repair in human paediatric surgery [11] and it has been shown to be superior compared to local anaesthetic infiltration for postoperative analgesia in children undergoing umbilical hernia repair [12].

There are some studies evaluating rectus sheath block in veterinary medicine. In 2022, Ienello et al. compared two volumes of injectate (low volume 0.5 mL kg^−1^, high volume 0.8 mL kg^−1^) administered cranially to the umbilicus in miniature swine [13]. The results showed that the injectate had spread mostly cranially, leaving the nerves caudal to the umbilicus free of stain. This was in contrast to our study, that used two injection points (cranial und caudal to the umbilicus), where we could see nerves staining in both directions. A study performed in dogs showed greater nerve staining when using a high volume (0.5 mL kg^−1^) compared to a low volume (0.25 mL kg^−1^) [14]. According to these results and due to a lack of a cadaveric study in foals evaluating volumes of spread, we decided to use 0.5 mL kg^−1^ per hemiabdomen divided between two injection points.

In 2023, Ishikawa et al. published a study evaluating a rectus sheath block first in two cadavers and afterwards in six healthy standing horses [19]. Although they used the same volume as in our study, they reported a higher proportion of nerves stained in the two cadavers (Th9 to L2) as opposed to our results (Th13 to L1). They also noticed that the dispersion of the injectate was limited to the rectus sheath, while in our study it reached the transversus abdominis muscle. This could be explained by several reasons. The injection point they employed was much more cranial that in our study, so more ventral branches of the thoracic spinal nerves could be reached. Second, the injectate in their study remained in the rectus sheath, therefore a greater cranio-caudal dispersion of the same volume of injectate could be achieved. Third, they included only adult horses in their study, which differ substantially in soft tissues composition, stiffness, tension, and anatomical conformation from the neonatal foals investigated in our study.

In 2021, Freitag et al. evaluated the influence of recumbency (lateral or dorsal) in an US-guided subcostal transversus abdominis plane block [15]. Analyzing the materials and methods in depth, the authors performed the injection of the contrast between the rectus abdominis muscle and the transversus abdominis. This approach is very similar to that performed in our study, except that both injection points were performed cranially to the umbilicus instead of cranially and caudally. The number of nerves stained in the dorsal recumbency group was very similar to our results, with the difference that they did not stain the most caudal nerves (Th18 and L1) as frequently as in the approach of the present study. That could be explained, as they mentioned in the discussion, because most of the nerves of the abdominal wall are situated in the area between the xyphoid cartilage and the umbilicus. Nevertheless, to provide complete analgesia of the periumbilical area, the nerves involved in the innervation of the most caudal part of the abdominal wall need to be reached (TH18 and L1) as shown in our results. The volume of the injectate used by Freitag and colleagues was much smaller than our study; nevertheless, the total number of nerves reached was similar. The explanation we have found is, as mentioned earlier, that they performed the block where most of the ventral branches of the thoracic spinal nerves are closely situated to each other.

As the internal layer of the rectus sheath is very thin in neonates, intraperitoneal injection can be possible. In our study, intraperitoneal dye was founded in 4/10 animals. As soon as we realised the high incidence of peritoneal injection, the injection technique was slightly modified. Administration of the injectate was started before the desired point was reached while advancing the needle, so that a “pocket” was created, and intraperitoneal injection was prevented. Nevertheless, under direct visualization, the probability of accidental organ puncture is low. There is a study comparing blind versus US-guided locoregional anaesthesia in cows that reported organ (renal and spleen) contamination with contrast media in the blind technique [16]. For this reason, and as previously mentioned, the authors prefer the US-guided over the blind technique [17]. It has to be considered that cadavers may suffer from autolytic processes. The anatomy could be altered, affecting the distribution of the injectate, and facilitating the contamination of the abdominal cavity.

One limitation of this study is the small number of cadavers employed. The number of animals included depended on the cadavers available during the time of the study. On the other hand, descriptive studies about locoregional techniques have usually employed a similar sample size [7,31]. The evaluation of only one volume of spread is also a limitation of the current study. A higher volume could result in a wider spread and consequently, a greater number of nerves stained. We used this volume based on preliminary results of pretrials and considering that this volume could be used in the clinical setting, bearing in mind the total dose of local anaesthetics, and attempting not to alter the anatomy of the surgical area. Another limitation is that injectate distribution in cadavers can differ from in vivo due to changes in muscle tension, temperature, absorption, motion, and blood and lymphatic flow. Finally, local anesthetics have a lower viscosity than the contrast agent used in the study, which may result in reduced distribution of the latter. 

## 5. Conclusions

The described US-guided RSB in foals is a feasible and reliable “in-plane” technique. The distribution of contrast in this study suggests that the block can provide analgesia for surgical procedures concerning the periumbilical area. Additionally, this approach avoids dorsal distribution of the injectate to the proximity of the femoral nerve, which could cause severe complications. However, further studies are necessary to evaluate different volumes of injectate and the extent of the block in living animals, as well as investigate its clinical efficacy.

## Figures and Tables

**Figure 1 animals-13-03600-f001:**
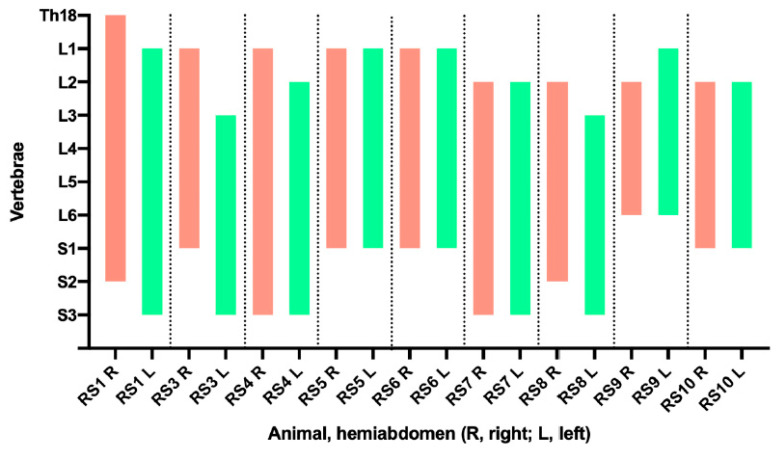
Graphical representation of the longitudinal extension of the contrast medium in relation to the vertebral bodies. Two perpendicular lines were drawn from the cranial and caudal limit of the contrast extent to the spine and the correspondent vertebrae were noted. Red represents the right hemiabdomen, green represents the left hemiabdomen. Th, thoracic; L, lumbar; S, sacral.

**Table 1 animals-13-03600-t001:** Evaluation of the ultrasound image quality during the realization of the block.

US Image Quality	Description
Excellent	Easily distinguishable shape of the triangle of RA and posterior leave of RAS. Separation of muscle layer and internal RS post-injection
Good	Identifiable muscle and peritoneum but no clear shape of the border of RAM
Poor	Landmarks could not be identified

**Table 2 animals-13-03600-t002:** Evaluation of the needle visualization during the realization of the block.

Needle Image Quality	Description
Excellent	Entire shape and tip identified
Good	Only tip visualized
Poor	Tip only identified after some fluid injection

**Table 3 animals-13-03600-t003:** Evaluation of the nerve staining.

Grade	Description
2	Complete, >1 cm and entire circumference
1	Partially, <1 cm or incomplete circumference
0	No staining, completely free of contrast

**Table 4 animals-13-03600-t004:** Animals, cranio-caudal extent of the contrast, and intraperitoneal and intramuscular contrast contamination. Data from carcase number 2 (RS2) could not be evaluated due to technical issues. Th, thoracic; L, lumbar; S, sacral; cm, centimeter.

Foal	Extent Injectate Right (cm)	Extent Injectate Left (cm)	Intramuscular Contamination	Intraperitoneal Contamination
RS1	17	16.8	NO	YES
RS2				
RS3	13.8	13.9	YES	NO
RS4	17.4	15.2	YES	YES
RS5	11.7	12.2	YES	YES
RS6	13.8	15	YES	NO
RS7	15.1	14.5	YES	NO
RS8	13.4	13.4	YES	NO
RS9	7.7	10.6	YES	NO
RS10	7.7	8.5	YES	NO

**Table 5 animals-13-03600-t005:** Number of nerves identified in the dissection, number and frequency of complete and partially stained nerves, and total frequency of nerves reached. Th, thoracic; L, Lumbar.

Spinal Nerve	Number of Nerves Identified	Completely Stained	Percentage of Complete Stained	Partially Stained	Percentage of Partially Stained	Total Percentage
Th13	20	2	10%	1	5%	15%
Th14	20	7	35%	1	5%	40%
Th15	20	13	65%	3	15%	80%
Th16	20	19	95%	0	0%	95%
Th17	20	17	85%	1	5%	90%
Th18	20	16	80%	1	5%	85%
L1	20	11	55%	1	5%	60%

**Table 6 animals-13-03600-t006:** Position of the umbilicus in relation to the spinal nerves. In most cases (70%) the umbilicus was situated between the ventral branches of the Th16 and Th17 spinal nerves. Th, thoracic.

Location	Number of Animals	Percentage
Th16-Th17	7/10	70%
Th17-Th18	3/10	30%

## Data Availability

Data is contained within the article or Appendix A.

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
