# Peer review of "Assessment of an Ultrasound-Guided Rectus Sheath Block in Foals: A Cadaveric Study"

_animals, 2023, doi:10.3390/ani13233600_

Round 1

Reviewer 1 Report

Comments and Suggestions for Authors

Dear Authors, 

Thank you very much for submitting your manuscript entitled 'Assessment of an ultrasound-guided rectus sheath block in foals: a cadaveric study.' to Animals. 

This manuscript is well-written and provides important information to the veterinary literature. Thanks for your contribution. 

I have included some minor comments/suggestions for the authors.

In the abstract (line 30) and along the manuscript, I suggest including nerve staining as an objective.

The study's main objective is to define the dye's arrival to the region's nerves. Therefore, this issue should be mentioned as the study's main outcome. The authors present it as the study's main conclusion even if it is not mentioned as an objective in the corresponding section.

Line 32. Please include in the abstract (if possible) the real injection point. Suggestion: 

Ten foal cadavers were included in the study, where a bilateral two-point ultrasound-guided rectus sheath block was performed with the injection divided into two separate aliquots: one administered 5 cm cranial to the umbilicus and the other 5 cm caudal to it.

Line 53. This sentence contains references to events relevant to humans and animals; however, it appears that Reference 7 pertains exclusively to animals. Please add a reference covering the human aspect or modify the sentence accordingly. 

Line 68. Thank you for providing the quote above. While it is certainly intriguing, I could not find any reference that addresses locoregional anesthesia's sparing effect in reducing inhalation agents. I may need to look at something, or rephrasing the sentence or including a relevant citation could be beneficial. I prefer the latter option, as I am unaware of any human studies confirming the role of blocks in reducing anesthetic agents.

Line 84. I suggest reformulating the objective mentioned nerve staining as an outcome and providing a hypothesis.

This study hypothesizes that a bilateral two-point ultrasound-guided rectus sheath injection at the umbilicus level in foals effectively reaches the ventral branches of the thoracic spinal nerves within the interfascial plane. To test this hypothesis, we assessed the feasibility of this injection technique in foal specimens. We evaluated the distribution pattern and nerve staining of a dye solution injected in the rectus sheath plane using computed tomography and gross anatomical dissection of foal carcasses.

Line 115. Please erase this sentence: "These animals were euthanized for causes unrelated to the study." 

This information has already been provided in the line 90.

Line 116. Please consider changing the sentence: "The study was performed within 24 hours after euthanasia. Before the experiment, the cadavers were stored in a cold room."

Line 147. Please consider changing "approach" for "technique". 

Both "in-plane technique" and "in-plane approach" can be appropriate when discussing regional anesthesia, but the choice between them depends on the context of your discussion.

. "In-plane technique" is commonly used to refer to the specific method or maneuver used to perform regional anesthesia, such as an in-plane ultrasound-guided technique.

. "In-plane approach" is more general and may refer to the overall strategy or approach used for regional anesthesia, considering factors like needle insertion and visualization.

So, if you are discussing a specific method or maneuver, "in-plane technique" might be more suitable. If you are discussing a broader approach or strategy, "in-plane approach" could be the better choice.

Line 175. The assumption of a linear relationship between distribution and volume may carry certain risks. It may be more prudent to limit our discussion to the extent achieved with the specific volume employed, noting that only one volume per point was used. The study was not explicitly designed to yield results on this specific point.

Please erase here and amend the rest of the manuscript and tables. 

Line 187. This sentence highlights that one of the study's objectives is to investigate this specific outcome (nerve staining).

Table 4. Please avoid using cm/mL as discussed above; the study was not explicitly designed to yield results on this specific point.

Line 199. Please refrain from including considerations or opinions within the results section. Additionally, it is important to note that fortune does not hold a place of prominence in scientific research.

"The CT scan results for the second animal (RS2) had to be excluded due to technical issues with the scanner." 

Table 4. Providing information about the number of vertebrae may be confusing, as the injection did not influence the lumbar nerve caudal ventral branches to L1 and sacral nerves. I propose reformulating how the authors present the extent of dye distribution in the CT image. You could represent it as a projection area where the nerves responsible for innervating the abdominal wall in that region are expected to be found. If this approach is not feasible, you can provide the size in centimeters of the observed stain.

Figure 1. The figure needs to be completed. No info can be visualized in it. I assume you are assessing the extent of the stain in relation to the spine. In that case, I recommend reviewing my previous comment for potential changes in how we present the data. Specifically, consider representing the extent as a projection area aligned with the nerves responsible for innervating the abdominal wall in that specific region. If this approach is not feasible, you can provide the size in centimeters of the observed stain.

Line 224. Please add the fact that the injection was divided into equal parts. 

"This is the first study that describes a four-point periumbilical ultrasound-guided rectus sheath block in foals with a total volume of 1 mL kg-1 divided into four equal aliquots."

Line 225. Please note that an objective not in the objectives section is presented here. 

Please change "entend" to "extent". 

The aim was to evaluate the extent of the dye and to quantify the number of spinal nerves reached. 

Line 294. Please add a period after "al."

Line 315. This underscores the challenge of assuming a linear relationship between the injected volume as cm/mL and the resulting distribution. Different volumes can yield similar distributions.

Line 321.  If modifications were made to the technique in 6 of the 10 animals based on contemporaneous findings during the study, it is crucial to specify this in the Materials section for transparency and clarity.

Line 329. There is a typo in the sentence. Please amend.

Line 345. Please change the "it is an easy" to "is feasible". Your study was not designed to evaluate the difficulty of the technique, and there are no results to support this conclusion.  

NOTE. As this innovative technique heavily relies on ultrasonography for precise guidance, I strongly recommend including a clear image of the region to be observed, emphasizing anatomical details prior to the injection. This would significantly enhance the understanding of the procedure in line with the references mentioned in the text (line 140). 

Once again, congratulations on your excellent work.

Best regards, 

Comments on the Quality of English Language

The text is well-written and easy to understand. 

Author Response

Dear Reviewer 1:

I appreciate a lot your review, the manuscript quality will be improved a lot with the corrections. Please find attached the corrections in the Word document. I copied your comments and below each one I wrote an answer. 

Best regards

Reviewer 2 Report

Comments and Suggestions for Authors

The manuscript describes the performance of an ultrasound guided rectus sheath block in foal cadavers. Subsequently,the distribution of stained nerves was assessed via CT and ultrasound.

The study is well-designed with reliable methods.methods and results are well-decribed and easy to follow. Results are appropriately discussed.

I have no more comments,beside:

Figure 1 does not show any graphs in my manuscripts, please check.

Please give a bit more explanation what exactly table 3 is displaying? (I guess innervation of the umbilicus)

Thanks for the work

Author Response

Dear Reviewer 2,

Thank you very much for taking time to read and review the manuscript. Please find attached a Word document with the answer to your comments. 

Do not hesitate to ask any question or to send any other comment you consider necessary. 
Best regards
